∂ | **Open Peer Review** | Environmental Microbiology | Research Article

# Seasonal dynamics and environmental controls of planktonic archaea in a typical subtropical estuary

Wenya Wei,[1] Penghui Li,[1] Fahui Gong,[2] Cheng Zhang,[1] Kedong Yin,[1] Wei Xie[1]

**ABSTRACT** Planktonic archaea are pivotal in the biogeochemical cycling across estuaries to coastal seas. A thorough comprehension of their adaptive mechanisms to seasonal environmental fluctuations remains largely unexplored. This study investigates the seasonal dynamics of planktonic archaeal communities and their responses to the biotic and abiotic factors in the Pearl River Estuary (PRE). Thermoproteota and Thermoplasmatota are the two most abundant phyla, and both show a significant difference between summer and winter. As phosphorus is the most limiting nutrient in the PRE, $PO_4^{3-}$ is found to have the most significant seasonal variation in random forest, followed by temperature and Chl a. The Mantel test for the abundant archaea showed that the number of phosphate-correlated OTUs was the second highest, following only temperature. The generalized additive modeling (GAM) analysis further reveals that the abundance of Thermoproteota was controlled by $PO_4^{3-}$, temperature, and DO, whereas MGII was controlled by $PO_4^{3-}$, pH, salinity, and Chl a. Our research demonstrates that there is a strong seasonality in coastal archaeal communities and sheds light on revealing their environmental adaptation and predicting biogeochemical function alterations in response to regional and global environmental changes.

**IMPORTANCE** Archaea not only sustain the equilibrium of elemental cycles but also exhibit remarkable plasticity in responding to and adapting to fluctuating environmental conditions. In particular, the adaptive strategies and ecological impacts of archaea in complex and dynamic settings, such as estuaries, represent a compelling yet unresolved area of scientific inquiry. Our study focused on the seasonal dynamics of planktonic archaeal communities in the Pearl River Estuary (PRE) and their response to biotic and abiotic factors. Our study demonstrates a strong seasonality in the aggregation of these coastal archaeal communities and adaptability to dynamic phosphate concentrations, emphasizing the critical role of phosphate in controlling the distribution of archaea. Our study sheds light on revealing environmental adaptation and predicting biogeochemical function alterations in response to regional and global environmental changes.

**KEYWORDS** subtropical estuary, planktonic archaea, seasonality, phosphorus, environmental adaptation

Archaea exhibit a pervasive presence across diverse natural habitats, encompassing terrestrial soils (1), marine (2, 3), and freshwater (4) systems, as well as extreme environments, such as deep-sea hydrothermal vents (5), hypersaline lakes (6), and permafrost zones (7). Their ubiquitous distribution underscores the pivotal role that archaea play in different ecosystems. The ecological functions of archaea, notably their contributions to the cycling of C, N, and P, are a testament to their indispensable influence on biogeochemical processes (8). These archaea not only sustain the equilibrium of elemental cycles but also exhibit remarkable plasticity in responding to and adapting to fluctuating environmental conditions. In particular, the adaptive strategies

**Peer Reviewers** Meiling Yang, Tianjin University, Tianjin, China; Changyi Zhang, University of Illinois Urbana-Champaign, Urbana, Illinois, USA

Address correspondence to Wei Xie, xiewei9@mail.sysu.edu.cn.

The authors declare no conflict of interest.

and ecological impacts of archaea in complex and dynamic settings, such as estuaries (9, 10), represent a compelling yet unresolved area of scientific inquiry. The main archaea groups in the estuaries are Thermoproteota (mainly composed of Marine Group I [MGI]), Thermoplasmatota (including Marine Group II, III [MGII, MGIII]), and Bathyarchaeota. In general, Bathyarchaeota occupy the sedimentary environments of most estuaries, while Thermoproteota and Thermoplasmatota were reported as the major groups in the water column.

Studies related to estuaries reveal that variations in nutrient concentrations (e.g., nitrogen and phosphorus) significantly influence microbial communities, which is attributed to the unique high nutrient inputs, water transport, and biodegradation in estuaries (11, 12). MGIs are autotrophic or mixotrophic and generate energy by oxidizing ammonium to nitrite or ammonia oxidation (13). Thus, MGIs are crucial in maintaining the balance of the global nitrogen cycle due to their high affinity for ammonium. Compared with the contributions of archaea in nitrogen cycling, the research about them in phosphorus cycling is still limited. A recent physiological experiment suggests that phosphate assimilation by MGIs is on the same order as annual riverine input of phosphate, and roughly 4% of that is consumed by photoautotrophs on the surface, demonstrating the importance of the MGIs in phosphorus cycling (8). Due to their prolonged adaptation to the unique conditions of estuarine ecosystems, archaea inhabiting these environments have developed adaptive mechanisms to cope with the fluctuating levels of phosphorus availability. For example, some planktonic archaea have been reported to secrete multiple enzymes that decompose organic phosphorus in phosphate-deficient environments, supporting their growth and metabolism (8). Several studies suggest that AOA can utilize different types of phosphate sources through additional encoded phosphate transporter protein genes and phosphorus capture systems. The high-affinity phosphate transporter system is retained in P-limited environments, while the low-affinity phosphate transporter system is lost to reduce the energy-consuming processes associated with DNA replication (14, 15). An estuarine MGII is also reported to contain a high-affinity phosphate transporter and thus might be more competitive in dynamic phosphate concentrations (16). Phosphorus usually has seasonal differences in estuarine environments, with seasonal runoff and rainfall flushing causing higher TP and DOP in the PRE during the wet season than dry season, while the dilution effect of high runoff results in lower DIP during the wet season (17).

Except for the phosphate, other biotic or abiotic factors also have effects on the distribution and ecological functions of those archaea in estuaries. For example, MGI is overall most abundant in the water column of the Charente and Brodkill estuaries, while MGII dominates in summer seawater (18). This has been explained by the fact that seasonal variability factors, i.e., temperature and photo-availability rate (PAR), strongly influence the composition and abundance of archaeal communities, while tides and salinity also strongly control the composition of archaeal communities (19). Besides environmental factors, the structure and dynamics of archaeal communities are also affected by biological interactions. Co-occurrence network analysis can provide a more comprehensive understanding of the assembly processes of the archaeal community and their responses to environmental changes (20, 21). However, few studies have examined the network properties of archaeal communities in the estuary under different seasons. Thus, more studies should be focused on understanding the adaptive mechanisms of archaeal communities in response to the dynamical estuary environments and characterizing their biogeochemical contribution.

The PRE, China's second-largest river system, has a typical subtropical monsoon climate and is influenced by both natural elements, such as ocean currents and tides, and anthropogenic activities. The dissolved inorganic N/P in this region fluctuated from 4.33 to 242.50 (with an average of 42.60 ± 47.06), which is far higher than Redfield ratio (16:1) (22), indicating that P limitation persists in the PRE (23). The average annual flux of phosphate discharged from the PRE was $3.34 \times 10$ mol/month, which accounted for 30.94% of the total phosphorus flux (17). The phosphate concentrations

also exhibit seasonal fluctuations, ranging from 0.56 µM during the summer months to 1.44 µM in the winter months (24). This variation encompasses a broad spectrum of phosphate availability for those planktonic archaea (8), suggesting that they may experience periods of phosphate limitation. The PRE thus could be considered as a natural laboratory for investigating the adaptive mechanisms of archaeal communities to the seasonal dynamic phosphate conditions. Despite those ecological significance, the seasonal dynamics of the archaeal community in this subtropical estuary have been sparsely investigated, particularly under varying phosphate conditions across different seasons. Therefore, the objectives of our study were to (1) clarify the compositions of archaeal communities (2); elucidate the spatiotemporal distribution patterns of archaeal communities; and (3) decipher the abiotic and biotic factors controlling the distribution and assembly of archaeal communities in the PRE in different seasons with contrasting phosphate concentration.

## MATERIALS AND METHODS

### Sample collection and environmental factor analysis

Surface (~1 m) and bottom water (8–50 m) samples were collected using rosette bottle samplers from 33 sites in July–August 2021 (termed summer samples) and 23 sites in December 2021–January 2022 (termed winter samples) (Fig. S1; Table S1). Water salinity, temperature, and dissolved oxygen (DO) were recorded by the CTD system. Each 2 L water was prefiltered with a 0.2 µm filter membrane (Whatman Nuclepore PC membrane). A total of 0.2 µm filters—64 in summer and 45 in winter—were obtained for microbial community analysis. Dissolved inorganic nutrients ($PO_4^{3-}$ $SiO_4^{2-}$, $NO_3^-$, $NO_2^-$, and $NH_4^+$) were analyzed using a SEAL autoanalyzer system. $PO_4^{3-}$ and $SiO_4^{2-}$ concentrations were determined by molybdenum blue spectrophotometry. Concentrations of $NO_2^-$ and $NO2_3^-$ were determined using diazotization-coupling spectrophotometry and cadmium column and spectrophotometry. The indigo phenol blue spectrophotometry was used to determine $NH_4^+$ concentration. Chlorophyll a (Chl a) on the glass fiber membrane after filtering the water sample was determined by spectrophotometry.

### DNA extraction, 16S rRNA gene amplification, and qPCR amplification

Those 0.2 µm filters were used for DNA extraction using the FastDNA Spin Kit for Soil (MP bio, USA), following the manufacturer's protocols. The archaeal 16S rRNA gene was amplified by PCR using primers Arch344F: 5′-ACGGGGYGCAGCAGGCGCGA- 3′ and Arch915R: 5′-GTGCTCCCCCGCCAATTCCT-3′ (25). The PCR amplification procedures were performed according to the manufacturer's standard protocol for TransStart FastPfu DNA Polymerase (TransGen Biotech, China). Amplification specificity was confirmed by a single peak in melt curve analysis (Tm: 57°C). Primer efficiency (>90%) was validated via standard curves ($R^2 > 0.99$) using serial dilutions of DNA. Cycle threshold (Ct) values ranged from 16 to 33 across samples. Three PCR replicates were performed for each sample DNA, and the PCR products from the three replicates were mixed. The products were detected by 2% agarose gel electrophoresis, recovered and purified using the AxyPrep DNA Gel Extraction Kit (Axygen Biosciences, USA), and quantified with a Quantus Fluorometer (Promega, USA). The samples were mixed according to the amount of sequencing required for each sample. The qPCR was performed on a Roche 480 machine using SYBR-Green PCR Master Mix (Takara, Kyoto, Japan). The DNA fragments of the 16S rRNA gene recovered from PCR products were used as templates for standard curves. It was prepared using a 10-fold series from 7.32E + 03 to 7.32E + 09 copies/µL for archaeal 16S rRNA genes. All reactions were performed in triplicate, and the specificity of q-PCR assays was verified by checking the dissociation curves and amplification curves.

## High-throughput sequencing and sequence data processing

The amplified products were high-throughput sequenced using a paired-end strategy on the Illumina MiSeq PE300 (Illumina Inc., San Diego, CA, USA) by Shanghai Meiji Bio-Pharmaceuticals Technology Company Limited (Shanghai, China). The raw sequenced sequences were filtered by fastp (https://github.com/OpenGene/fastp, version 0.19.6) for quality control (26) and spliced by FLASH (http://www.cbcb.umd.edu/ software/flash, version 1.2.7) (27). Then, the sequences were noise-reduced using DADA2 built in the QIIME2 standard pipeline (version 2020.6) (28). The retained high-quality sequences were clustered into operational taxonomic units (OTUs) at a 97% sequence similarity threshold. Species annotations for representative sequences were obtained by comparing them to the SILVA database (release 132) using a 70% comparison threshold (29).

## Statistical analysis

Spearman's correlation analysis and the "diversity" function of the "vegan" package to analyze the contributions of the main environmental variables of the archaeal alpha diversity index (Richness, Shannon, Simpson, Pielou, and ACE) and species abundance. Random forests were constructed using the "randomForest" package of R, and the mean decrease accuracy was used to determine the importance of the features in the initial categorization. The Mantel test calculated Pearson's correlations between environmental variables and archaeal OTUs by using the "linkET" package in R. Redundancy analysis (RDA) was performed to visualize the relationship between archaeal community composition and explanatory environmental variables using the Canoco5 software and the "corrplot" package in R.

Co-occurrence networks analyze species coexistence by examining the correlation of species abundance across different samples. OTU tables with a total relative abundance above 0.005 were filtered out, and the Spearman's correlations and $p$-values were calculated using the "Hmisc" and "reshape2" package in R. OTUs with a correlation coefficient > |0.6| and a $P$-value < 0.05 were selected and imported into the Gephi (v0.9.7). The archaeal co-occurrence network was visualized using the Fruchterman-Reingold algorithm, and the network metrics—including average degree, clustering coefficient, network density, average connectivity, average path length, betweenness centrality, and modularity—were all calculated in Gephi.

## Model establishment

The species distribution model is a research model that statistically fits different factors to quantitative features, such as organism occurrence and abundance, to reveal the correlation between the distribution and abundance of species and factors and is widely used in environmental ecology research. In this study, the abundance of archaea and major taxa was used as the dependent variable, and temperature, salinity, pH, DO, Chl a, $PO_4^{3-}$, $SiO_4^{2-}$, $NH_4^+$, $NO_2^-$, and $NO_3^-$ were used as the explanatory variables. The variance inflation factor (VIF) was applied to test the multiple covariance of the influencing factors separately to screen the factors suitable for inclusion in the model, and the factors with VIF >5 were removed before modeling. The generalized additive modeling (GAM) model was constructed using the "mgcv" package. The abundance of planktonic archaea was log-transformed (LnA) to conform to a normal distribution. The expression of the GAM model is:

$$\mathrm{LnY} = \alpha + s(x1) + \ldots + s(xn) + \varepsilon$$

where $Ln$ is a link function, $Y$ is a response variable, $s()$ denotes the smoothing function, $x$ is the independent variable, and $\varepsilon$ is the error term.

The stepwise regression method was utilized to filter the factors during model construction, and the Akaike Information Criterion (AIC) was followed to select the dependent variables and build the optimal model. The smaller the AIC and the larger

the cumulative variance explained ratio, the better the model fitting effect. Generalized cross-validation (GCV) and $R^2$ were used to evaluate the model fit; a higher $R^2$ and the lower GCV indicate a better-constructed model. $p$-value indicates the significance of the statistical results.

## RESULTS

### The seasonal physicochemical parameters of sampling sites

The physicochemical properties of the water column were quantified to delineate the environmental conditions conducive to the survival of archaea. Except for temperature, which exhibited a mean of 26.6 ± 2.7°C during the summer and 19.5 ± 0.7°C in the winter months, the most statistically significant differences ($P < 0.001$) in environmental parameters between the two seasons were observed for $PO_4^{3-}$ concentrations, with winter values averaging 1.29 ± 0.69 µM and summer values at 0.4 ± 0.2 µM, chlorophyll a (Chl a) levels, which were 0.57 ± 0.18 µg/L in winter and 3.52 ± 2.45 µg/L in summer, and dissolved oxygen (DO), with winter concentrations at 6.21 ± 0.31 mg/L and summer concentrations at 4.67 ± 0.99 mg/L (Fig. S3). Additionally, salinity and pH exhibited significant seasonal variations ($P < 0.05$). The other tested parameters, including $NH_4^+$, $SiO_4^{2-}$, and $NO_3^-$, show no significant difference ($P > 0.05$) between the two seasons (Fig. S3).

### Archaeal community structure and composition

To determine the diversity of the archaeal community, high-throughput sequencing was performed using archaea-specific primer 344F/915R. Overall, 7,057 archaeal OTUs were identified from 2,342,643 high-quality sequences at a 95% similarity level for the 109 water samples after trimming and chimera removal. The archaeal community included nine phyla and four superphyla. The most diverse of OTUs is Nanoarchaeaeota, while the highest OTU abundance is Thermoproteota (55.13%) and shared more than 80% of the archaeal community with Thermoplasmatota (28.72%). The relative abundance of archaeal taxa varied across the season (Fig. 1). The most abundant phylum was the Thermoproteota, followed by the Thermoplasmatota in the summer and winter, with the difference that the Thermoproteota abundance was considerably higher in the winter (68.36%) than in the summer (45.75%). At the level of the class, the richness and diversity were higher in summer, with Nitrososphaeria (MGI, 45.74%) and Poseidoniia (MGII, 32.29%) being the dominant species, while in winter, Nitrosopumilaceae (68.35%) was overwhelmingly dominant in the total archaeal taxa. In addition, archaeal community structure did not differ significantly between water layers, either in summer or winter.

### Alpha diversity of the archaeal communities

The alpha diversity was calculated for each season at the OTU level, and statistically different factors are displayed in Fig. 2A. Both the richness index (Richness and ACE) and the diversity indices (Shannon, Simpson, and Pielou) are higher in summer than in winter; however, only the diversity index is significantly higher in summer than in winter. The rank abundance curve of the OTUs in summer and winter samples is plotted in Fig. S2. The width of the curve reflects species richness, and the shape (smoothness) of the curve reflects species evenness. The summer samples have a greater range on the horizontal axis and a smoother curve, indicating greater species richness and a more even distribution of species than in winter.

### Patterns and driving factors of archaeal community composition

The differences between environmental factors in the summer and winter samples were calculated by the Wilcoxon signed-rank test. Among all significant differences, salinity, DO, $PO_4^{3-}$, and $NO_2^-$ were higher in winter than in summer, while higher temperature, pH, and Chl a concentration were higher in summer than in winter. Other environmental

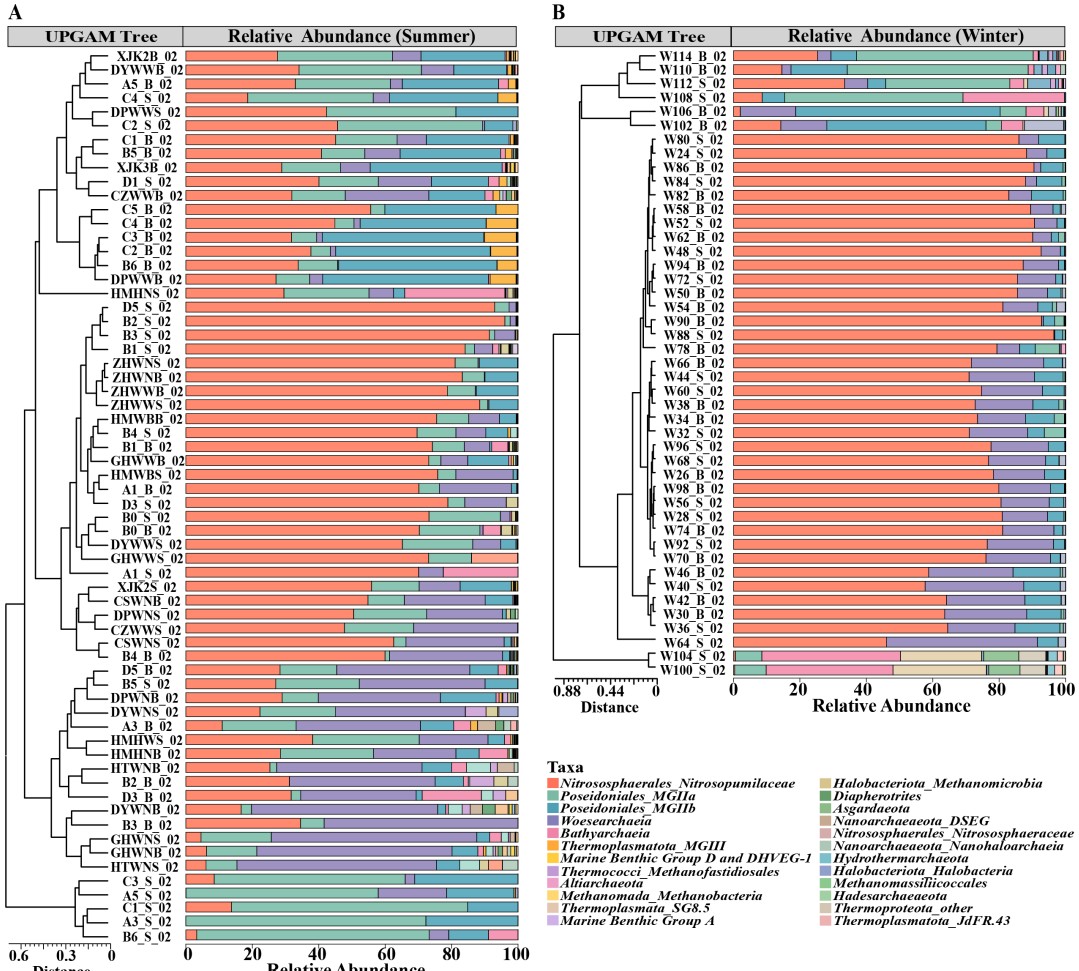

**FIG 1** Community composition of archaea from 109 seawater samples in summer (A) and winter (B). The most relative abundances of archaea were shown at class level. The UPGMA tree construction was generated using the Bary-Curtis corrected genetic distance to calculate pairwise distances between OTUs derived from 16S rRNA gene sequences (bootstrap = 1000).

factors did not differ significantly between seasons (Fig. S3). Furthermore, during the training of the random forest (RF) model, the contribution of each feature to the predictive performance was assessed through its mean decrease accuracy (MDA), which measures the importance of a variable for classification. In this study, 10 environmental factors were used with a classification tree set to 500. The significant features based on random forest indicated that phosphate, temperature, and Chl a were the key factors contributing to seasonal differences ($P < 0.001$) (Fig. 3A). The Mantel test examined OTU responses to environmental variables to assess the extent to which they are affected by environmental variables. The top 50 archaeal OTUs by relative abundance accounted for 78% of the total community and were mainly composed of Nitrosopumilaceae and MGII. The number of phosphate-correlated OTUs (12/50) was the second highest, following only temperature (17/50) (Fig. 3B). Mantel's test at the family level similarly showed that Nitrosopumilaceae (MGI) abundance was significantly affected by temperature and phosphate. Meanwhile, MGIIa abundance was significantly correlated with temperature, while MGIIb was significantly correlated with temperature, salinity, and chlorophyll a (Fig. S4B). The RDA showed that the environment of the community composition in the winter samples was characterized by higher DO, $PO_4^{3-}$, $SiO_2^-$, $NH_4^+$, $NO_2^-$, and $NO_3^-$, and lower temperature, Chl a, and pH compared to the summer samples (Fig. S4A). At the class level, RDA1 and RDA2 explained 24.92% and 3.66% of the variance. Nitrososphaeria (Nitrosopumilaceae) showed significant positive correlations with $PO_4^{3-}$ ($P < 0.01$), while

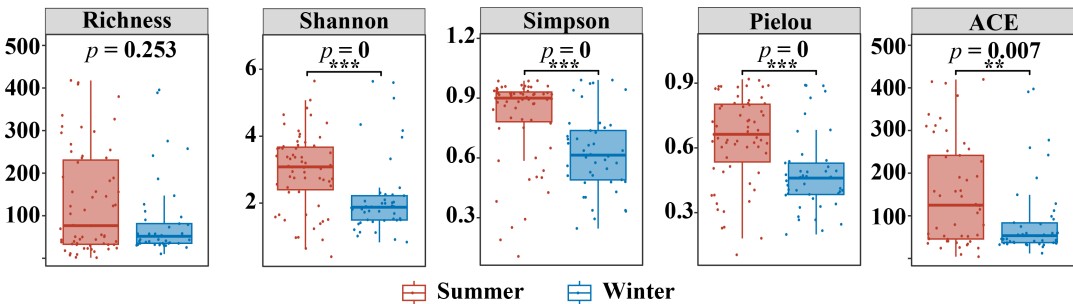

**FIG 2** The differences in archaeal community α diversity index between summer and winter. The horizontal bar represented the median number, and the error bars indicated the standard deviation (SD). (*$P < 0.05$, **$P < 0.01$, ***$P < 0.001$).

Thermoplasmatota, including MGIIa, MGIIb, MGIII, and Marine Benthic Group D and DHVEG-1 (MBGD and DHVEG-1), showed significant positive correlations with pH ($P < 0.01$) and significant negative correlations with $SiO_4^{2-}$ ($P < 0.01$). Woesearchaeia and Bathyarchaeia had a significant positive correlation with temperature ($P < 0.01$)(Table S2).

## Co-occurrence network affecting the microbial community composition

Based on the hierarchical cluster analysis of Bray-Curtis distances, the samples were divided into high- and low-salinity groups (Fig. S5). The co-occurrence network was performed using samples from the high-salinity group. The result showed that co-occurrence networks of archaea between summer and winter, and the network of the winter was more complex than in summer due to more edges, although there were fewer nodes than in summer (Summer: 368 nodes, 2,701 edges; Winter: 241 nodes, 4,380 edges). The average path length in winter is shorter than that in summer, and the average connectivity is higher than that in summer, indicating that winter co-occurrence networks were more stable and complex than summer networks (Table S3). The dominant species groups were roughly the same in both networks, with the four most frequently occurring taxa being Woesearchaeia, Bathyarchaeia, MGIIa, and Nitrosopumilaceae. Bathyarchaeia, Marine Benthic Group D and DHVEG-1, and Asgardaeota more frequently appeared in winter's network (Fig. 4).

## Model performance and effect of the individual predictor on archaeal abundance

The qPCR results for 16S rRNA genes of archaea are presented in Fig. S6. In winter, the abundance of archaea ranged from $6.40 \times 10^5$ to $5.21 \times 10^8$ copies $L^{-1}$. In summer, the abundance varied between $7.03 \times 10^3$ and $8.07 \times 10^6$ copies $L^{-1}$. Specifically, at the PRE, the abundance of archaea was significantly higher in winter than in summer.

Multiple covariance tests were applied individually to the influencing factors using VIF, and factors, including temperature, salinity, pH, DO, $NH_4^+$, $PO_4^{3-}$, and Chl a concentration, could be added to the model (Table S4). The optimal expression of the GAM model was obtained according to the criteria (higher $R^2$ values and lower AIC values) in Table S5, and the final expression of the GAM model for archaeal abundance copy number with environmental variables is shown in Table 1.

The results showed that the variables that most affected the archaeal abundance in our study were $PO_4^{3-}$, temperature, and DO. The abundance of archaea increased slowly with the increase of $PO_4^{3-}$ and DO concentration and decreased with the increase in temperature. The trends for Nitrosopumilaceae abundance were similar to archaea. MGIIa positively correlated with $PO_4^{3-}$ concentration, salinity, and pH, while MGIIb was additionally negatively correlated with Chl a concentration on this basis. MGIII abundance showed a negative correlation with temperature, DO concentration, and Chl a concentration, and a positive correlation with $PO_4^{3-}$ concentration and pH. The abundance of Bathyarchaeia, MBGD and DHVEG-1, methanofastidiosaceae, and

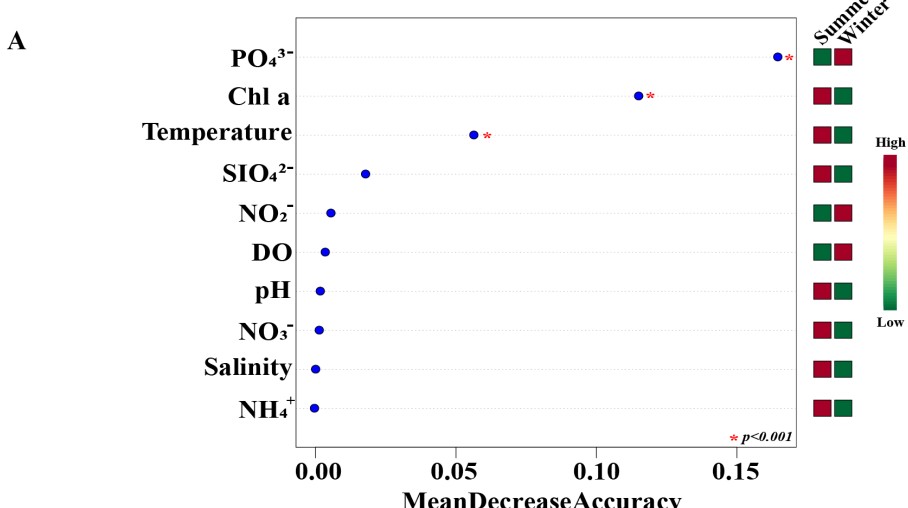

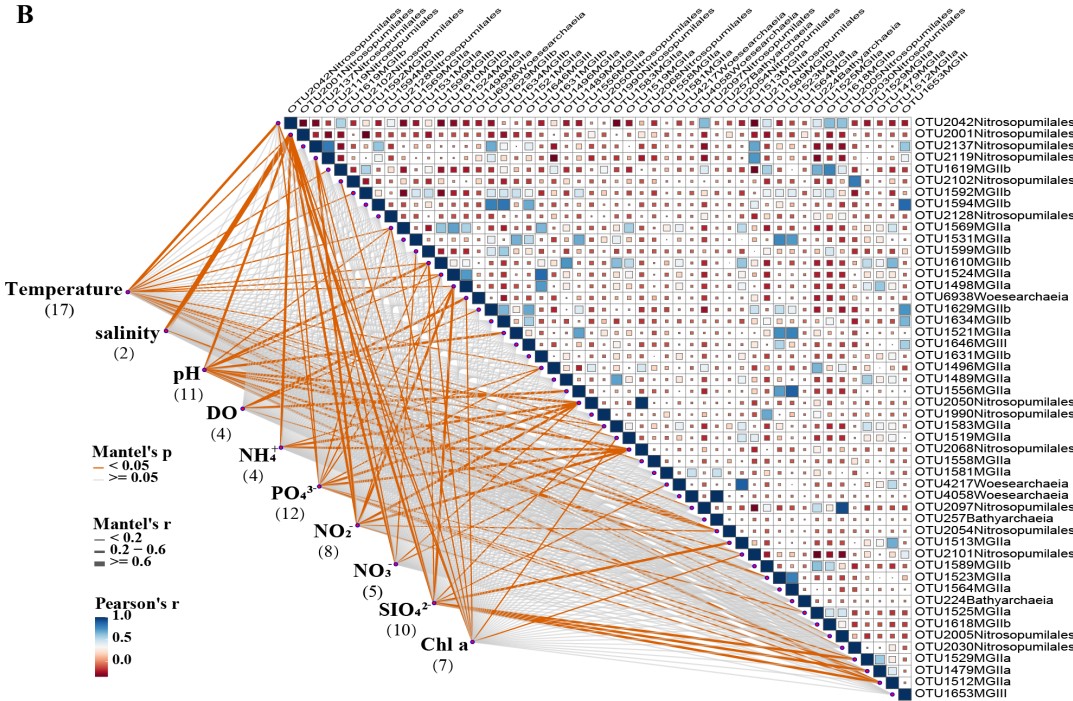

**FIG 3** Random forest analysis (A) and Mantel test analysis (B) for the environmental drivers on the archaea OTUs. The pairwise relationship of OTUs is calculated by the Pearson correlation and represented by color gradients. The edge width represents Mantel's r statistic, and the edge color represents statistical significance. The numbers in parentheses below the environment variable represent the number of OTUs significantly correlated with it.

Asgardaeota decreases with increasing temperature, salinity, and DO concentration. Meanwhile, Asgardaeota also has a positive correlation with pH, and methanofastidio-sales has a negative correlation with pH. The Woesearchaeia abundance is positively correlated with $PO_4^{3-}$ concentration and is negatively correlated with temperature and salinity (Fig. 5).

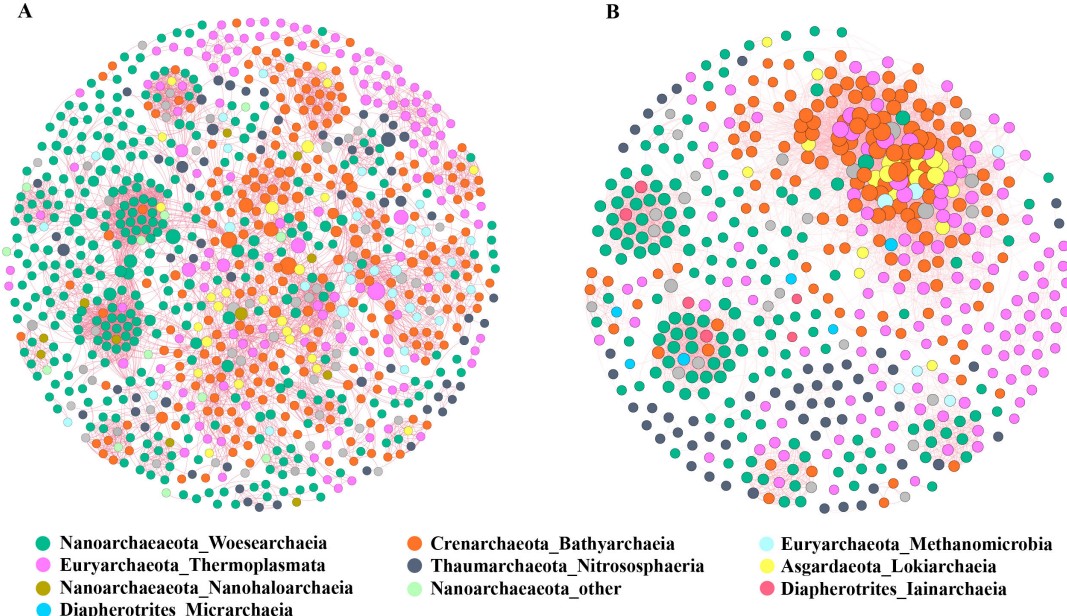

**FIG 4** Co-occurrence networks of the archaeal communities with a Spearman's coefficient > |0.6| and a *P*-value < 0.05 between OTUs in summer (A) and winter (B). The red lines represent significant positive correlations, and the green lines represent significant negative correlations. A node represented an OTU, and the node size was proportional to the number of connections.

## DISCUSSION

### Seasonal variation shapes archaeal community diversity and composition

The Pearl River estuary (PRE) has a typical subtropical monsoon climate, where both temporal and spatial dynamics drive diverse archaeal ecological niches (30). Seasonal variations lead to significant changes in the physicochemical parameters of PRE, such as temperature, salinity, DO, $PO_4^{3-}$, and Chl a concentration. The high phosphate concentration in the PRE in winter is due to stronger vertical mixing of seawater, which suspends phosphate from the sediments (31), as well as massive dead phytoplankton decomposition in winter, converting DOP to phosphate. In summer, the freshwater dilution and poor mixing of the water column due to the thermocline prevent bottom phosphate from reaching the surface layer, while large amounts of phytoplankton utilization of phosphate resulted in lower phosphate levels (24). The seasonal differences in this study are mainly due to the mixing water of tidal and freshwater inputs driven by winter monsoon, as demonstrated in previous studies (32). Higher temperatures allow fuller

**TABLE 1** Performance of species distribution model fitting[a]

| Taxa | Model |
| --- | --- |
| Archaea | LnArchaea = s(Temperature)+s(DO)+s($PO_4^{3-}$)+ε |
| Asgardaeota | LnAsgardaeota = s(Temperature)+s(Salinity)+s(pH)+s(DO)+ε |
| Bathyarchaeia | LnBathyarchaeia = s(Temperature)+s(Salinity)+s(DO)+ε |
| MBGD and DHVEG-1 | LnMBGD and DHVEG-1 = s(Temperature)+s(Salinity)+s(DO)+ε |
| Methanofastidiosaceae | Ln Methanofastidiosaceae = s(Temperature)+s(Salinity)+s(pH)+s(DO)+ε |
| MGIIa | LnMGIIa = s(Salinity)+s(pH)+s($PO_4^{3-}$)+ε |
| MGIIb | LnMGIIb = s(Salinity)+s(pH)+s($PO_4^{3-}$)+s(Chl a)+ε |
| MGIII | LnMGIII = s(Temperature)+s(pH)+s(DO)+s($PO_4^{3-}$)+s(Chl a)+ε |
| Nitrosopumilaceae | LnNitrosopumilaceae = s(Temperature)+s(DO)+s($PO_4^{3-}$)+ε |
| Woesearchaeia | LnWoesearchaeia = s(Temperature)+s(Salinity)+ε |

[a]The ln represent the logarithm of archaeal abundance; s(Temperature) is water temperature; s(Salinity) is salinity; s(pH) is pH; s(Chl a) is chlorophyll a concentration; s(DO) is dissolved oxygen concentration; s($NH_4^+$) is ammonium concentration; and s($PO_4^{3-}$) is phosphate concentration.

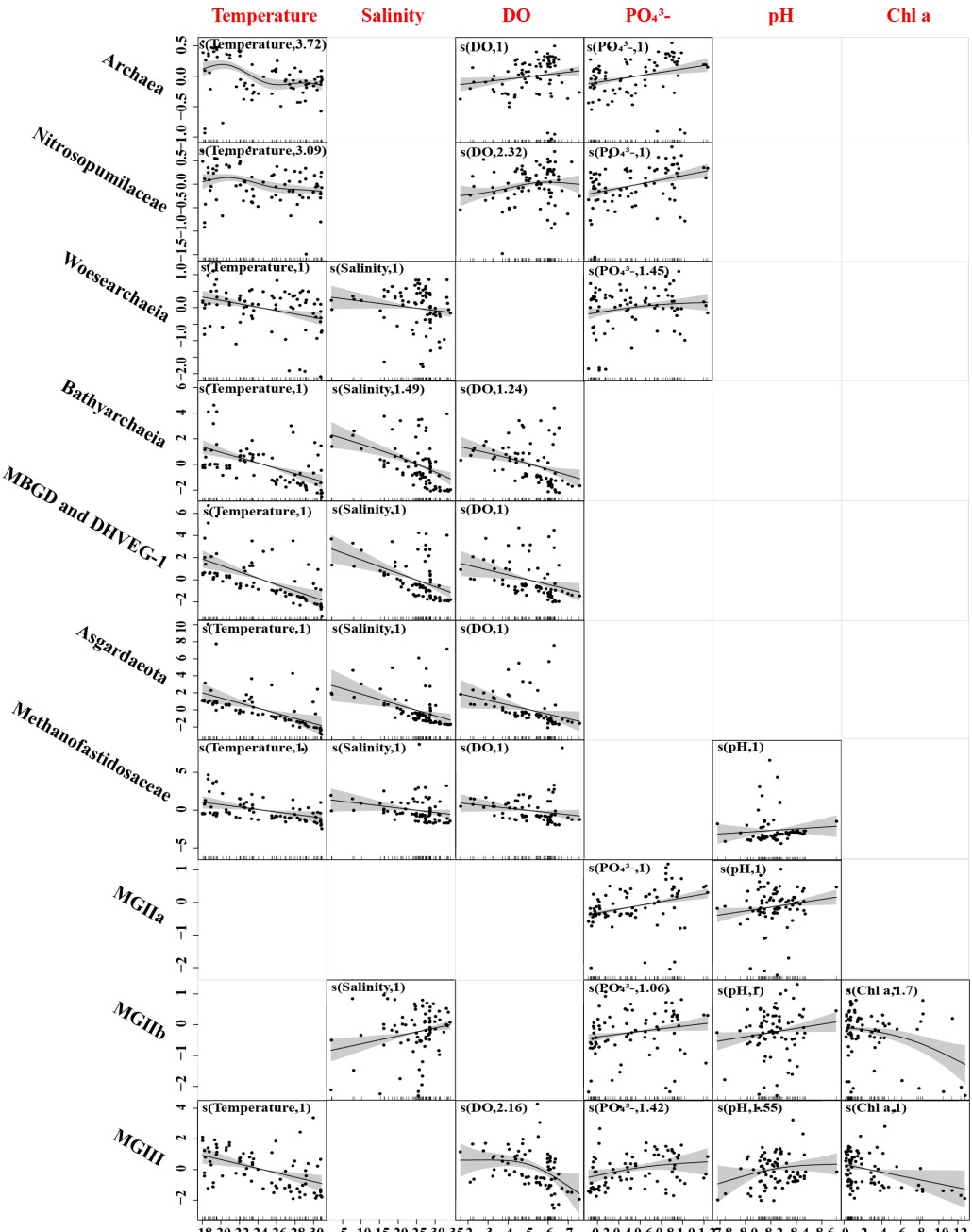

**FIG 5** Contributions of environmental factors to archaea and major taxa abundance in GAM. The contributions of different environmental factors to archaea and major taxa abundance are plotted on the y-axis. The number on the y-axis labels represent the effective degree of freedom. Solid lines with significant relationships ($P < 0.01$) represent smoothing lines from GAMs, and shadow indicates a 95% confidence interval of the estimation. The inward tick marks on the X-axis show data distributions.

photosynthesis by seawater phytoplankton, which in turn leads to higher chlorophyll a concentrations in summer seawater. The DO concentrations (mean 6.17 mg/L) were observed to be higher in the winter samples of this study than in the summer.

Variation in environmental parameters due to seasons shapes the composition and diversity of microbial communities (2), which are characteristics of estuarine ecosystems globally (33–35). Overall, our study demonstrates that Thermoproteota is the main archaeal community in the shallow seawaters of the PRE, followed by Thermoplasmatota, similar to previous studies (36). However, the distribution of archaeal-dominant taxa

varied slightly between seasons. The relative abundances of Thermoproteota increased significantly in winter, mainly in Nitrososphaeria. Variations in AOA abundance may be influenced by changing environments, such as dissolved oxygen, and higher dissolved oxygen concentrations in winter samples can stimulate ammonia-oxidation processes, which is a critical process in the marine nitrogen cycle (37). To survive in the phosphates-limited estuary, AOA needs to exhibit a higher affinity for phosphate to increase the efficiency of phosphorus absorption, which may explain the high abundance of AOA in high-phosphate winters. The MGII abundance increased significantly during summer, probably because its heterotrophic metabolism and light-trapping capacity make it more likely to occur in warmer waters (38). The diversity and richness of archaea were significantly higher in summer than in winter. Higher temperatures can promote microbial growth and activity by altering rates of enzyme activity and membrane transport efficiency (39). Seasonal shifts in environmental factors due to seasonal variations were more influential in determining archaeal abundance than seawater layers and lifestyles.

## The responses of archaeal communities and their abundance to abiotic factors

Previous studies showed that temperature, salinity, and nutrient concentrations are key factors in shaping the specific community structure of estuarine archaea in the PRE (2). The Mantel test showed that $PO_4^{3-}$ was most significantly correlated with high abundance OTUs (Nitrosopumilaceae), followed by temperature and DO, implying the criticality of $PO_4^{3-}$. In phosphorus-limited environments, archaea enhance the expression of high-affinity phosphate transporter proteins, such as PstB, to more efficiently capture phosphate from the environment. In phosphorus-sufficient environments, archaea may rely more on low-affinity phosphate transport systems, such as Pit, to reduce energy consumption (14). Nitrosopumilaceae belongs to autotrophic archaea (AOA) and is considered to be able to convert ammonium to nitrate using oxygen (40). The high abundance distribution of AOA as drivers of the key nitrification rate-limiting step implies that the low phosphate utilization and high nitrification rates were adaptive responses to phosphorus limitation in the PRE water column. Additionally, some heterotrophic archaea (Woesearchaeia, Bathyarchaeia, and Methanofastidiosaceae) are strongly correlated with temperature, pH, and salinity, factors that also influence the growth of Thermoplasmatota (MGIIa, MGIIb, and MGIII) in this study. Environmental factors and archaeal composition also had a clear seasonal distribution in RDA. The random forest showed that phosphate had the most significant effects on the seasonal variation, followed by temperature and Chl a. Subtropical estuaries are likely to exhibit strong seasonal variation in environmental conditions (41–43), and archaea may have similar ecological responses to environmental changes allowing them to occupy specific ecological niches. Seasonal variation leads to shifts in dominant taxa, with estuarine AOA being more adaptable and performs stronger functions than AOB during the cold season (44). Overall, temperature, $PO_4^{3-}$, DO, and pH may be the main factors contributing to the seasonal variation of the archaeal community in PRE.

GAM, a widely used statistical framework, has been extensively applied to examine the influence of environmental factors on biological abundance within aquatic ecosystems. GAM is distinguished by its ability to model a response variable as a sum of smooth functions of predictor variables, prioritizing the maximization of fit smoothness. This approach quantitatively evaluates the impact of environmental parameters on microbial abundance and projects microbial biomass shifts in the context of climate change scenarios. In marine and lacustrine environments, GAM has been instrumental in forecasting the biomass and distribution patterns of both flora (45) and bacteria (46). However, the application of GAM to predict alterations in the archaeal communities of estuaries across varying environmental conditions remains largely unexplored. This gap in the literature underscores the need for further research to harness the predictive

capabilities of GAM in understanding the dynamics of archaeal populations in estuarine settings.

The environmental preferences of the archaea can be analyzed through the collection of environmental parameters in the marine water, and the spatial and temporal distribution of the archaea could provide important support for the adaptation to climate change. The GAM model revealed that $PO_4^{3-}$ was the most powerful environmental factor affecting the abundance of these dominant archaea (MGI and MGII accounted for 73.43% of the total abundance). The phosphate concentration in our study was significantly higher in winter (1.130 ± 1.069 µmol/L) than in summer (0.427 ± 0.263 µmol/L) seawater, which may be due to the dilution effect of high runoff caused by high rainfall in summer (47). Based on Deutsch, N* ($[NO_3^-]$ - 16 × $[PO_4^{3-}]$ + 2.9 mmol/L) in seawater was utilized to assess the bias in the Redfield ratio, and our study suggests that the degree of phosphorus limitation is relieved in winter compared to summer (48). With the highest copy number in winter, AOA likely possess a unique phosphorus acquisition mechanism that enables them to survive and adapt to phosphorus-limited environments. Several studies have demonstrated the presence of high-affinity phosphate transporters, inorganic phosphorus-dependent regulators, or polyphosphate utilization (ppA) genes in the MAGs of AOA, which allow them to acquire phosphorus in the surrounding seawater environment (49). Compared to low-phosphate seawater, AOA uses the phosphate transporter capacity to capture more phosphate easily in high-phosphate environments, which is also consistent with our study that AOA has a higher abundance in high-phosphate environments in winter. This phenomenon also suggests that Nitrosopumilaceae tends to dominate in environments with high phosphate bioavailability. Similar to Nitrosopumilaceae, we found that both MGIIa and MGIIb were positively correlated with phosphate. The superiority of MGII abundance in the PRE may be related to its special phosphate-limited environment. Compared with MGII in other environments, MGII in PRE—with a high N:P ratio (~100:1)—contains a high-affinity phosphate transporter and thus might be more competitive (16), indicating the significant effects of phosphate on MGII.

Studies have pointed out that oxygen is critical in controlling the distribution and activity of marine AOA (50). Our results similarly suggest that AOA can live in a wider range of DO concentrations, which is an indication of its adaptation to the environment. Warmer seawater absorbs less oxygen, and heterotrophic archaea may prefer to live in nutrient-rich, low-oxygen environments. Temperature directly affects the metabolic activity of archaea as well as most biochemical pathways. As the dominant order in the PRE, Nitrosopumilaceae abundance increases at temperatures below 20°C but decreases gradually as temperatures rise >20°C, which also explains their greater abundance during the colder winter months.

Our study showed a rising trend in MGIIb with increasing salinity, and this trend weakened at salinities higher than 25 PSU. In general, MGIIb was mostly detected in oligotrophic and high-salt seawaters. The strong correlation between MGII abundance and salinity may be due to its brackish origin and the evolution in different salinity water environments (51). Besides, salinity was one of the main factors affecting the heterotrophic archaea abundance. This effect depends on different subgroups, such as Bathy-8, which was the richest relative abundance in the marine environment, and has a highly significant positive correlation with salinity, while Bathy-6 has a significant negative correlation with salinity (52).

The MGIIb and MGIII were the only species in this study that were negatively correlated with Chl a, which can be explained by its widespread presence in the deep sea with low chlorophyll environments. MGIII was also more prevalent in environments with low temperatures and dissolved oxygen.

Interestingly, our modeling shows no association of archaeal abundance with ammonium, possibly related to the special ecosystem of the PRE. High mobility and multiple sources of nitrogen in the PRE resulted in a higher nitrogen growth rate than phosphate (phosphate limitation). Moreover, inorganic nitrogen was dominated

by nitrate, followed by nitrite, with ammonia at the lowest, so the abundance of archaea in the surface water of the PRE was less affected by ammonium.

## Deciphering archaeal interactions with co-occurrence networks

Co-occurrence networks reveal complex interactions within microbial communities (53). Previous studies reported the coexistence of archaea and bacteria with eukaryotes, whereas geographic latitude and season influence the complexity and stability of estuarine archaeal communities (54). In this study, positive correlation associations dominated the co-occurrence network of archaea, suggesting a potential symbiotic relationship between archaea. The co-occurrence network had more links in winter despite more nodes in summer, which suggests a more complex network in winter.

Bathyarchaeia and Asgardaeota were key taxa in winter high-mixing seawater and were closely related in the winter network, suggesting that they prefer symbiosis in oligotrophic environments. Bathyarchaeota is involved in mineralized organic matter, acetogenesis, methanogenesis, urea production, and other processes (55, 56). In turn, Asgard can utilize its products for biochemical processes, which contributes to their cooperative or symbiotic relationship. Besides, members of the Bathyarchaeia and the Thermoplasmatota often co-occur in multiple environments (57). The higher nodes and connectivity of them in this study demonstrate their integral role in maintaining network stability. Nutrients are more readily available to microorganisms in the summer, and archaea do not require more symbiotic relationships to obtain organic matter. This may explain why the Thermoplasmatota, Bathyarchaeia, and Woesearchaeia have more nodes and connections in the summer, but fewer connections.

As the most abundant archaeal taxon, Nitrososphaeria was not predominant in the co-occurrence networks of either winter or summer, unlike in previous studies (58). Meanwhile, Nitrososphaeria is not closely connected to other species in the co-occurrence network, implying that its community structure is not much influenced by biotic factors. It suggests that the abundance and co-occurrence characteristics of archaea are not necessarily synchronized. Although seasonal changes were insignificant for the major taxa of archaea, they altered the average connectivity between communities. Overall, the network topology points to a more tightly and complexly connected archaeal community in winter and is more stable to external environmental disturbances.

## Conclusions

Our comprehensive investigation into the spatial and temporal distribution patterns of archaeal communities revealed that the phyla Thermoproteota and Thermoplasmatota dominate the archaeal assemblages. The pronounced seasonal fluctuations significantly influence the distributional dynamics and community structure of these archaea. The seasonal variations in abiotic factors, especially the phosphate, DO, pH, salinity, and Chl a concentrations, play pivotal roles in controlling the abundances of those abundant archaea, such as Nitrosopumilaceae, MGIIa, and MGIIb. Co-occurrence networks show a more tightly and complexly connected archaeal community in winter than in summer, suggesting the importance of biotic factors in shaping the archaeal community. Overall, our study presents the adaptation of those coastal archaea to the dynamic phosphate concentration and provides insight into predicting their prospective changes and biogeochemical functions in the context of global environmental changes.

## ACKNOWLEDGMENTS

We thank the captain and crews of R/V "Hai Si Lu Liu Hao" during the 2021 winter and 2022 summer cruises. We also thank Pan Huang for the help with the sampling and filtering work for this study. We thank Junjian Liang, Xiaomin Wang, Xinmei Wang, Guowei Bao, Xinlin Liang, Zhiqiao Chen, and Weiwen Ye for their assistance with this experiment during the summer cruise, and Jinzheng Chen, Xiaomin Wang, Qinglin Yang, Changjie Wang, and Zuoyan Zheng for their assistance with this experiment during the winter cruise.

This work was supported by the National Key Basic Research Program of China (2022YFC2805505, Wei Xie); the Project of Southern Marine Science and Engineering Guangdong Laboratory (Zhuhai) (SML2023SP218, SML2021SP204, SML2023SP215, SML2023SP238, Wei Xie, Kedong Yin); the National Natural Science Foundation of China (92051117, 41776137, Wei Xie); the Guangdong Basic and Applied Basic Research Foundation (2021B1515120080, 2023A1515012149, Wei Xie, Fahui Gong); and the Program of Department of Natural Resources of Guangdong Province (GDNRC[2021]62, Kedong Yin).

Wenya Wei: Conceptualization, Data curation, Formal analysis, Investigation, Methodology, Visualization, Writing–original draft, Writing–review and editing. Penghui Li: Investigation, Writing–review and editing. Fahui Gong: Resources, Writing–review and editing. Cheng Zhang: Resources, Writing–review and editing. Kedong Yin: Funding acquisition, Investigation, Project administration, Resources, Writing–review and editing. Wei Xie: Conceptualization, Funding acquisition, Investigation, Methodology, Project administration, Resources, Supervision, Writing–original draft, Writing–review and editing.

## AUTHOR AFFILIATIONS

[1]School of Marine Sciences, Sun Yat-sen University & Southern Marine Science and Engineering Guangdong Laboratory (Zhuhai), Zhuhai, China
[2]Key Laboratory of Coastal Environmental Processes and Ecological Remediation, Yantai Institute of Coastal Zone Research, Chinese Academy of Sciences, Yantai, China

## AUTHOR ORCIDs

Wenya Wei http://orcid.org/0000-0001-8041-4577
Wei Xie http://orcid.org/0000-0003-4508-3041

## AUTHOR CONTRIBUTIONS

Wenya Wei, Conceptualization, Data curation, Formal analysis, Investigation, Methodology, Validation, Writing – original draft, Writing – review and editing | Penghui Li, Investigation, Writing – review and editing | Fahui Gong, Resources, Writing – review and editing | Cheng Zhang, Resources, Writing – review and editing | Kedong Yin, Funding acquisition, Investigation, Project administration, Resources, Writing – review and editing | Wei Xie, Conceptualization, Funding acquisition, Investigation, Methodology, Project administration, Resources, Supervision, Writing – original draft, Writing – review and editing

## DATA AVAILABILITY

The data sets presented in this study can be found in online repositories. The names of the repository/repositories and accession number(s) can be found below: https://www.ncbi.nlm.nih.gov/bioproject/PRJNA1146731

## ADDITIONAL FILES

The following material is available online.

### Supplemental Material

**Figure S1 (Spectrum00759-25-s0001.tif).** Maps of the study area and sampling sites for summer (A) and winter (B) in Pearl River Estuary and adjacent coastal area.
**Figure S2 (Spectrum00759-25-s0002.tif).** Abundance rank curves based on OTU levels in winter and summer of the Pearl River Estuary.
**Figure S3 (Spectrum00759-25-s0003.tif).** Difference of environmental factors in summer and winter in the Pearl River Estuary.

**Figure S4 (Spectrum00759-25-s0004.tif).** Redundancy analysis (RDA) for the relation-ship between archaeal community composition and environmental parameters. Red arrows, environmental factors; blue arrows, species; blue squares, winter samples; yellow circles, summer samples.

**Figure S5 (Spectrum00759-25-s0005.tif).** Hierarchical clustering analysis of 16S rRNA sequences at the OTU level based on Bray-Curtis differences (blue, high salinity group; red, low salinity group).

**Figure S6 (Spectrum00759-25-s0006.tif).** Archaeal abundance copies/L (log10) of 16S rRNA gene in summer and winter. (A) 16S rRNA gene copies/L in summer; (B) 16S rRNA gene copies/L of archaeal dominant taxa in summer; (C) 16S rRNA gene copies/L in winter; (D) 16S rRNA gene copies/L of archaeal dominant taxa in winter.

**Supplemental tables (Spectrum00759-25-s0007.csv).** Tables S1 to S5.

## Open Peer Review

**PEER REVIEW HISTORY (review-history.pdf).** An accounting of the reviewer comments and feedback.

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
