## [Reviewer comments · Microbiology Spectrum]

Microbiology Spectrum

Seasonal Dynamics and Environmental Controls of Planktonic Archaea in a Typical Subtropical Estuary

wen wei, peng Li, fa Gong, cheng zhang, Kedong Yin, and Wei Xie

Corresponding Author(s): Wei Xie, Sun Yat-Sen University

Review Timeline:

Submission Date:	March 12, 2025
Editorial Decision:	June 30, 2025
Revision Received:	August 11, 2025
Accepted:	September 29, 2025

Editor: Gabriella Fiorentino

Reviewer(s): Disclosure of reviewer identity is with reference to reviewer comments included in decision letter(s). The following individuals involved in review of your submission have agreed to reveal their identity: meiling yang (Reviewer #1); Changyi Zhang (Reviewer #2)

Transaction Report:

DOI: <https://doi.org/10.1128/spectrum.00759-25>

Re: Spectrum00759-25 (**Seasonal Dynamics of Planktonic Archaea in a Typical Subtropical Estuary: Implications for Their Phosphate Adaptation**)

Dear Prof. Wei Xie:

Thank you for the privilege of reviewing your work. Below you will find my comments, instructions from the Spectrum editorial office, and the reviewer comments.

Revision Guidelines

Sincerely,
Gabriella Fiorentino
Editor
Microbiology Spectrum

Reviewer #1 (Comments for the Author):

This study employed relatively reasonable experimental techniques and analytical methods.

Reviewer #2 (Comments for the Author):

The manuscript by Wei et al. presents a comprehensive and well-executed study investigating the seasonal dynamics of planktonic archaeal communities in the Pearl River Estuary (PRE). By using a multifaceted approach, including high-throughput 16S rRNA gene sequencing, qPCR, co-occurrence network analysis, and generalized additive models (GAMs), the authors convincingly demonstrate that phosphate concentration likely plays a central role in structuring archaeal distribution patterns. This integrative work offers a solid framework for exploring environmental drivers of archaeal diversity and abundance, and it significantly contributes to our understanding of archaeal ecology in dynamic estuarine environments.

In general, the manuscript is well written, methodologically sound, and rich in data. The experimental procedures are clearly described, and the results are thoroughly analyzed and thoughtfully interpreted. Particularly, the integration of GAM modeling with microbial data is a valuable approach. I essentially have no big concerns except a few minor suggestions for clarification and improvement.

Specific Comments:

Lines 148-152: Please provide additional details regarding the qPCR assay, including Ct values, amplification specificity (e.g., melt curve analysis), and primer efficiency. Also, consider whether "RT-PCR" should be corrected to "qPCR" in this context. It seems that the reverse transcription was not part of the assay.

Line 164: Please justify why a 70% threshold was selected for taxonomic assignment.

Lines 226-227: Since OTU identification was performed using a specific archaeal 16S primer set, have the authors considered how different primer choices might influence the community composition results? A brief discussion of potential primer bias would enhance the methodological transparency.

Lines 242-243: Please specify the method used to generate the UPGMA tree (e.g., distance metric, clustering algorithm). This information could be included in the methods section or as part of the figure legend.

Lines 313-315: Clarify the number of technical and/or biological replicates used in the qPCR assays. This information is important for evaluating the reliability and reproducibility of the quantitative data.

The theme of the article is “Seasonal Dynamics of Planktonic Archaea in a Typical Subtropical Estuary: Implications for Their Phosphate Adaptation”. Although the author has endeavored to analyze the aspect of phosphorus limitation, the evidence regarding the relationship between archaea and phosphorus is still somewhat tenuous or insufficient. On the contrary, the analysis of the co-occurrence and other abiotic factors such as water temperature, dissolved oxygen, and biotic factors on the community structure is more sufficient. Therefore, I believe that the current analysis results cannot support the existing topic. It is hoped that more sufficient evidence regarding how phosphorus affects the abundance of archaea and the structure of their communities can be added, or the content can be reorganized to change the topic.

There are still some issues that need to be further revised or pondered over:

1 The figure and text do not match. See details from the line 216 to the line 223.

2 From Fig. S2, it can be observed that chlorophyll a (chl a) exhibits the most pronounced seasonal variations compared to phosphate. Microalgae and archaea interact through both synergy and competition in material cycles, such as microalgae synthesizing organic carbon available for

archaea, while both compete for resources like dissolved organic carbon (DOC) and ammonium (NH_4^+). Therefore, theoretically, chl a is a more critical determinant of archaeal abundance and community structure. However, this aspect was not thoroughly explored in this study.

3 Fig. 1 should be plotted for summer and winter seasons respectively, which would make it easier to compare the differences in community structure.

4 In Fig. 3, in winter, phosphorus is the main influencing factor for the community structure; in summer, water temperature and Chl a are the most significant influencing factors. However, in Lin269 and Line270, the differential influences of phosphorus, water temperature and Chl a were ignored.

5 In Fig.3 , the connections between each environmental factor and the archaeal OTUs are extremely dense, making it impossible to clearly distinguish the key influencing factors for each taxon. Should you consider using other taxonomic levels instead of OTUs for Mantel test analysis?

6 Based on Fig. S1, S5 and Fig. 2, there are significant seasonal differences in the microbial community structure. It is suggested that Fig. 1 be split and described and discussed separately.

7 Fig. S5 shows obvious seasonal differences rather than salinity clustering. Why was only co-occurrence analysis for high salinity

conducted?

8 Check line 325, s(Chl a)

9 In lines 327-339, Fig. 5 should be added.

10 For line 432-433, how to determine the phosphorus-limited environment of archaea? Why are AOA still in a phosphorus-limited state when the phosphorus concentration in water bodies is significantly higher in winter than in summer?

11 In line 441-446, the text emphasizes that phosphorus might be a significant influencing factor for MGIIa and MGIIb. However, in Fig. S4, it is quite clear that pH and salinity are the key influencing factors. How can this be explained?

Response to Reviewer Comments

Dear Editors and Reviewers:

Thank you for your letter and for the reviewers' comments concerning our manuscript entitled "Seasonal Dynamics of Planktonic Archaea in a Typical Subtropical Estuary: Implications for Their Phosphate Adaptation". Those comments are invaluable and have greatly assisted us in revising and enhancing our paper, providing significant guidance for our ongoing research endeavors. We have carefully examined the comments and implemented the necessary corrections, which we believe address the concerns raised. Revised portions are marked in red in the paper. The main corrections in the paper and the responses to the reviewer's comments are as follows:

Reviewer #1:

Although the author has endeavored to analyze the aspect of phosphorus limitation, the evidence regarding the relationship between archaea and phosphorus is still somewhat tenuous or insufficient. On the contrary, the analysis of the co-occurrence and other abiotic factors such as water temperature, dissolved oxygen, and biotic factors on the community structure is more sufficient. Therefore, I believe that the current analysis results cannot support the existing topic. It is hoped that more sufficient evidence regarding how phosphorus affects the abundance of archaea and the structure of their communities can be added, or the content can be reorganized to change the topic.

Response:

We sincerely thank these critical points regarding our conclusion about the criticality of phosphorus for archaea. We agree that our current evidence is insufficient to support our former topic about the phosphate adaptation of those archaea. We then modified the title to "Seasonal Dynamics and Environmental Controls of Planktonic Archaea in a Typical Subtropical Estuary" to better align with the main focus of the article. In terms of the structure of the archaeal community, the Mantel test shows that the number of phosphate-correlated OTUs was the second highest, only after temperature. In terms of archaea abundance, the Generalized Additive Modeling (GAM) analysis further reveals that the abundance of Thermoproteota was controlled by PO_4^{3-} ($p=1.43\text{e-}06$), temperature ($p=8.75\text{e-}05$), and DO ($p=0.00615$). Besides, MGIIa was controlled by PO_4^{3-} ($p=1.36\text{e-}06$), pH ($p=0.0117$), and salinity ($p=0.0120$), and MGIIb was controlled by Chl a ($p=0.000909$), PO_4^{3-} ($p=0.001296$), salinity ($p=0.002049$), and pH ($p=0.028851$) (The Significance Coefficients of the environmental factors were added to TableS5 of supplementary table). Those statistical results demonstrated that phosphate is a critical environmental control on the distributions and abundance of those planktonic archaeal communities in the Pearl River estuary. Other factors, such as temperature, Chl a, DO, and salinity are also significant environmental controls on either distributions or abundance of those archaea, which have been fully considered in the revised version (temperature: (Results: Line 275-277/Line 284-285, Discussion: Line 469-473), Chl a (Results: Line 275-277/Line 284-285, Discussion: Line 483-486), DO (Results: Line 337-339, Discussion: Line 465-469), and salinity (Results: Line 339-342, Discussion: Line

474-482)) .

Response:

1. *Response to Comments: “The figure and text do not match. See details from the line216 to the line223.”*

Response: Thanks for your comments. The mistake has been modified in revised version (Line 224-226).

2. *Response to Comments: “From Fig. S2, it can be observed that chlorophyll a (chl a) exhibits the most pronounced seasonal variations compared to phosphate. Microalgae and archaea interact through both synergy and competition in material cycles, such as microalgae synthesizing organic carbon available for archaea, while both compete for resources like dissolved organic carbon (DOC) and ammonium (NH₄⁺). Therefore, theoretically, chl a is a more critical determinant of archaeal abundance and community structure. However, this aspect was not thoroughly explored in this study.”*

Response: We thank the reviewers for emphasizing the seasonal role of Chl a. In our study, higher Chl a concentrations had an effect on archaeal abundance (Mantel test) but were not a limiting factor for archaea. In addition, Chl a was lower in winter and did not significantly affect MGI abundance (GAM model). It appears that phosphate affects archaeal communities more broadly. Besides, the effect of chlorophyll a on archaea abundance(such as MGIIb and MGIII) was showed in GAM model (Result: Line 339-342; Discussion: Line 477-480).

3. *Response to Comments: “Fig. 1 should be plotted for summer and winter seasons respectively, which would make it easier to compare the differences in community structure.”*

Response: Thanks for your professional and helpful comments. We have plotted summer and winter separately and updated them in the revised version (Line 244).

4. *Response to Comments: “In Fig. 3, in winter, phosphorus is the main influencing factor for the community structure; in summer, water temperature and Chl a are the most significant influencing factors. However, in Lin269 and Line270, the differential influences of phosphorus, water temperature and Chl a were ignored.”*

Response: Thanks for your comments. We regret that the phrasing in this section caused confusion. The Random forests provide a rank of the importance of environmental factors that lead to seasonal differences in all samples and cannot explain changes in community structure within each season, so we only describe the rank of environmental factors importance in the main text. Meanwhile, seasonal effects on archaeal communities were shown in RDA (Figure S4A), with phosphate and DO positively correlated with MGI abundance in winter, then temperature, salinity, Chl a, and pH were positively correlated with MGII and MGIII in summer (Line 288-292).

5. *Response to Comments: “In Fig.3, the connections between each environmental factor and the archaeal OTUs are extremely dense, making it impossible to clearly distinguish the key influencing factors for each taxon. Should you consider using other taxonomic levels instead*

of OTUs for Mantel test analysis?”

Response: Thanks for your comments. We also did a Mantel test analysis of highly abundant archaeal communities with environmental factors at the family level. The results also indicate that, apart from temperature, phosphate significantly influences the abundance of the dominant species (MGI). The result has been added to Figure S4B and shown in the revised version (Line 282-285).

6. Response to Comments: “Based on Fig. S1, S5 and Fig. 2, there are significant seasonal differences in the microbial community structure. It is suggested that Fig. 1 be split and described and discussed separately.”

Response: Thanks for your comments. We have plotted summer and winter separately and updated them in the revised version (Line 244).

7. Response to Comments: “Fig. S5 shows obvious seasonal differences rather than salinity clustering. Why was only co-occurrence analysis for high salinity conducted?”

Response: Thanks for your comments. The low-salinity samples originate from the upper estuary, where strong and variable terrestrial inputs introduce substantial non-salinity-related variance and potential confounding factors. To reduce the influence of a few low-salinity samples upstream, we specifically analyzed high-salinity samples to identify seasonally driven patterns of co-occurrence.

8. Response to Comments: “Check line 325, s(Chl a)”

Response: Thanks for your comments. The information has been modified in revised version (Line 335).

9. Response to Comments: “In lines 327-339, Fig. 5 should be added.”

Response: Thanks for your comments. We have modified it in the revised version (line 349).

10. Response to Comments: “For line 432-433, how to determine the phosphorus-limited environment of archaea? Why are AOA still in a phosphorus-limited state when the phosphorus concentration in water bodies is significantly higher in winter than in summer?”

Response: Thanks for your comments. We didn't define the extent of the phosphorus limitation in this article, but we referenced the article by Deutsch (2007), which used N^* ($[\text{NO}_3^-] - 16 \times [\text{PO}_4^{3-}] + 2.9$ in mmol L^{-1}) in seawater to assess deviations from Redfield ratios. Positive N^* values usually indicate excess NO_3^- from nitrification and nitrogen fixation or other sources. In this study, the summer samples had an N^* value of 15.41, and the winter samples had an N^* value of 0.36, suggesting that excess NO_3^- was present in PRE. Compared to summer, this condition is relieved in winter but is still a phosphorus limitation. And, we have added this parameter in the revised version (lines 447-449).

[1] Deutsch C, Sarmiento JL, Sigman DM, Gruber N, Dunne JP. Spatial coupling of nitrogen inputs and losses in the ocean. *Nature*. 2007 Jan 11;445(7124):163-7.

11. Response to Comments: “In line441-446, the text emphasizes that phosphorus might be a significant influencing factor for MGIIa and MGIIb. However, in Fig. S4, it is quite clear that pH and salinity are the key influencing factors. How can this be explained?”

Response: Thanks for pointing this out. The apparent difference arises because the analysis in lines 441-446 is based on archaeal abundance (qPCR gene copy numbers), while Figure S4 is based on archaeal community composition (16S rRNA sequencing data). Phosphorus correlated significantly with the population size (abundance) of MGII subgroups, whereas pH and salinity primarily shaped the compositional structure of the broader archaeal community. We have made this more clearly in the revised version (Line 285, 410).

Reviewer #2:

The manuscript by Wei et al. presents a comprehensive and well-executed study investigating the seasonal dynamics of planktonic archaeal communities in the Pearl River Estuary (PRE). By using a multifaceted approach, including high-throughput 16S rRNA gene sequencing, qPCR, co-occurrence network analysis, and generalized additive models (GAMs), the authors convincingly demonstrate that phosphate concentration likely plays a central role in structuring archaeal distribution patterns. This integrative work offers a solid framework for exploring environmental drivers of archaeal diversity and abundance, and it significantly contributes to our understanding of archaeal ecology in dynamic estuarine environments.

In general, the manuscript is well written, methodologically sound, and rich in data. The experimental procedures are clearly described, and the results are thoroughly analyzed and thoughtfully interpreted. Particularly, the integration of GAM modeling with microbial data is a valuable approach. I essentially have no big concerns except a few minor suggestions for clarification and improvement.

Response: We are sincerely grateful to the reviewer for their thoughtful assessment of our work and their generous acknowledgment of its contributions. We deeply appreciate the positive feedback regarding the methodological rigor, data integration (particularly the GAM modeling), and the significance of our findings for archaeal ecology in estuarine systems.

Response:

1. Response to Comments: “Lines 148-152: Please provide additional details regarding the qPCR assay, including Ct values, amplification specificity (e.g., melt curve analysis), and primer efficiency. Also, consider whether “RT-PCR” should be corrected to “qPCR” in this context. It seems that the reverse transcription was not part of the assay.”

Response: Thanks for your comments. We confirm that “RT-PCR” was used erroneously and should be corrected to “qPCR” throughout (no reverse transcription step was involved) (Line 150, 154).

The requested details have been added to the revised manuscript (Section 2.2: Line142-145):

Ct values: All samples fell within the linear range (Ct: 16-33 cycles).

Specificity: Melt curve analysis showed single sharp peaks (Tm: 57°C), confirming primer specificity for archaeal 16S rRNA genes.

Primer efficiency: Standard curves (10-fold serial dilutions) for primers 344F/915R have a standard curve efficiency of >90% ($R^2 > 0.99$).

2. Response to Comments: “Line 164: Please justify why a 70% threshold was selected for taxonomic assignment.”

Response: Thanks for your important methodological comments. Higher thresholds (e.g., $\geq 90\%$) increase precision but discard $>30\%$ of archaeal reads, severely reducing statistical power; Lower thresholds (e.g., $\leq 50\%$) retain more sequences but risk substantial false-positive genus assignments. The 70% confidence threshold was chosen to balance the trade-off between classification precision and sequence retention, and selected based on:

(1) SILVA database recommendations for balancing classification precision and sequence retention (<https://www.arb-silva.de/documentation/release-138/>);

(2) Common practice in marine microbiome studies^[1] to minimize false genus-level assignments while retaining $>80\%$ of quality-filtered reads.

[1] Parada A E, Needham D M, Fuhrman J A. Every base matters: assessing small subunit rRNA primers for marine microbiomes with mock communities, time series and global field samples. *Environmental microbiology*, 2016, 18(5): 1403-1414.

3. Response to Comments: “Lines 226-227: Since OTU identification was performed using a specific archaeal 16S primer set, have the authors considered how different primer choices might influence the community composition results? A brief discussion of potential primer bias would enhance the methodological transparency.”

Response: Thanks for your comments. The archaeal-specific primers 344F/915R were selected for two key reasons:

(1) Field Standardization: This primer pair is the most widely adopted for marine/estuarine archaeal surveys, enabling direct comparison with global datasets;

(2) High specificity: This primer pair is designed to amplify archaeal 16S rRNA genes with minimal cross-amplification of archaeal sequences, ensuring a more accurate representation of archaeal communities.

4. Response to Comments: “Lines 242-243: Please specify the method used to generate the UPGMA tree (e.g., distance metric, clustering algorithm). This information could be included in the methods section or as part of the figure legend.”

Response: Thanks for your comments. The UPGMA tree construction in this article was generated using the Bary-Curtis corrected genetic distance to calculate pairwise distances between OTUs derived from 16S rRNA gene sequences. And the tree was constructed using the unweighted pair-group method with arithmetic means, implemented in MEGA 12. We performed 1,000 bootstrap replicates to assess the robustness of the tree topology. These details have been added to the figure legend (Figure 1) in the revised version (Line 245-248).

5. Response to Comments: “Lines 313-315: Clarify the number of technical and/or biological replicates used in the qPCR assays. This information is important for evaluating the

reliability and reproducibility of the quantitative data.”

Response: Thanks for your comments. Each qPCR reaction was performed in triplicate to minimize technical variability. The experiments were conducted using 109 independent seawater samples collected in summer (33 sites) and winter (23 sites) to account for natural environmental variation.

Re: Spectrum00759-25R1 (**Seasonal Dynamics and Environmental Controls of Planktonic Archaea in aTypical Subtropical Estuary**)

Dear Prof. Wei Xie:

Your manuscript has been accepted, and I am forwarding it to the ASM production staff for publication. Your paper will first be checked to make sure all elements meet the technical requirements. ASM staff will contact you if anything needs to be revised before copyediting and production can begin. Otherwise, you will be notified when your proofs are ready to be viewed.

Sincerely,
Gabriella Fiorentino
Editor
Microbiology Spectrum

Reviewer #2 (Comments for the Author):

The authors have made substantial revisions in line with the reviewers' comments. My concerns have been fully addressed, and I have no additional questions